# Efficient Lignin Fractionation from Scots Pine (*Pinus sylvestris*) Using Ammonium-Based Protic Ionic Liquid: Process Optimization and Characterization of Recovered Lignin

**DOI:** 10.3390/polym14214637

**Published:** 2022-10-31

**Authors:** Sharib Khan, Daniel Rauber, Sabarathinam Shanmugam, Christopher W. M. Kay, Alar Konist, Timo Kikas

**Affiliations:** 1Biosystems Engineering, Institute of Forestry and Engineering, Estonian University of Life Sciences, 56, Kreutzwaldi, 51006 Tartu, Estonia; 2Department of Chemistry, Saarland University, Campus B2.2, 66123 Saarbrücken, Germany; 3London Centre for Nanotechnology, University College London, 17-19 Gordon Street, London WC1H 0AH, UK; 4Department of Energy Technology, Tallinn University of Technology, Ehitajate tee 5, 19086 Tallinn, Estonia

**Keywords:** renewable biomass, lignin, protic ionic liquid (PIL), sustainable biomass processing, lignin extraction, depolymerization

## Abstract

Lignin-based chemicals and biomaterials will be feasible alternatives to their fossil-fuel-based counterparts once their breakdown into constituents is economically viable. The existing commercial market for lignin remains limited due to its complex heterogenous structure and lack of extraction/depolymerization techniques. Hence, in the present study, a novel low-cost ammonium-based protic ionic liquid (PIL), 2-hydroxyethyl ammonium lactate [N11H(2OH)][LAC], is used for the selective fractionation and improved extraction of lignin from Scots pine (*Pinus sylvestris*) softwood biomass (PWB). The optimization of three process parameters, viz., the incubation time, temperature, and biomass:PIL (BM:PIL) ratio, was performed to determine the best pretreatment conditions for lignin extraction. Under the optimal pretreatment conditions (180 °C, 3 h, and 1:3 BM:PIL ratio), [N11H(2OH)][LAC] yielded 61% delignification with a lignin recovery of 56%; the cellulose content of the recovered pulp was approximately 45%. Further, the biochemical composition of the recovered lignin and pulp was determined and the recovered lignin was characterized using ^1^H–^13^C heteronuclear single quantum coherence (HSQC) nuclear magnetic resonance (NMR) spectroscopy, quantitative ^31^P NMR, gel permeation chromatography (GPC), attenuated total reflectance (ATF)–Fourier transform infrared spectroscopy (ATR-FTIR), and thermal gravimetric analysis (TGA) analysis. Our results reveal that [N11H(2OH)][LAC] is significantly involved in the cleavage of predominant *β–O–4*’ linkages for the generation of aromatic monomers followed by the in situ depolymerization of PWB lignin. The simultaneous extraction and depolymerization of PWB lignin favors the utilization of recalcitrant pine biomass as feedstock for biorefinery schemes.

## 1. Introduction

As a progressive society, the human race has thrived on the unrestrained usage of natural resources, especially fossil resources, to generate fuels and chemicals. However, the extensive exploitation of fossil fuels creates paramount pollution stress over our planet’s environment, which aids in global warming. Therefore, new strategies/techniques need to be designed to identify sustainable alternatives for the production of fuels and building blocks for chemicals from renewable carbon sources [1]. Lignocellulosic biomass (LCB), a bio-renewable substrate consisting of cellulose (44%), hemicellulose (30%), and lignin (26%) with an annual global production of about 181.5 billion tons, holds potential as a sustainable replacement of fossil resources [2]. Among the LCB constituents, lignin is a major component; however, it has had lower commercial interest than cellulose. On the other hand, lignin is the only natural polymer containing aromatic phenylpropanoid monomers, which makes it an exciting candidate for producing fuels, chemicals, and polymers [3].

Lignin is an amorphous heteropolymer that typically consists of aryl ether linkages (*β-O-4*) which constitute <50%, followed by *β–β* and *β–5* carbon linkages. The oxidative disruption of inter-unit linkages and the subsequent oxidation of hydroxyl groups results in lignin degradation to generate high-value, low-molecular-weight aromatic compounds [4]. Due to its complex heterogenic structure, recalcitrance, and lack of techniques for depolymerization, it is predominantly burned to meet energy requirements [5]. Currently, 50 million tons/year of lignin is generated from the Kraft process, whereas only 2% is commercially used [6]. To efficiently depolymerize lignin, an efficient technique that extracts lignin in its natural form is required. The most commonly used methods employ energy-intensive harsh reaction conditions, which tend to modify lignin’s structure [7].

Recently, ionic-liquid (IL)-mediated lignin extraction has emerged as an environmentally benign technique for the selective fractionation of various types of biomass [8]. ILs are salts (with a melting point of <100 °C) that are liquid at ambient temperatures and comprise an organic cation and an organic/inorganic anion in their structures. The infinite combinations of cations and anions with their unique properties (high thermal and chemical stabilities, chemical tunability, nonflammability, and higher ionic conductivity) make it possible to design ILs as “tailor-made” solvents for specific applications [9]. Interestingly, the presence of acidic protons in ILs structures make them potential candidates for the selective fractionation/oxidation of lignin. Based on the availability of protons on the cation, the ILs are distinguished into protic (PIL) and aprotic (APL) ionic liquids. Compared with APLs, PILs have several advantages in terms of cost-effectiveness, easy synthesis, availability of one or more H^+^ in the cation, high recyclability, and low toxicity. Due to the availability of more protons on the cations, PILs are considered ideal candidates for lignin extraction [10]. Moreover, the pretreatment of a lignocellulosic biomass using PILs offers the advantage of simultaneous extraction and activation (increment of carbonyl groups) of lignin, which can be used to synthesize hybrid materials [11].

The economic sustainability of ionic-liquid-mediated pretreatment depends upon the optimization of the process parameters for the enhanced selective fractionation of biomass. Specifically, the higher biomass-loading ratio intensifies the pretreatment and decreases the capital expenditure of the process [12]. Hence, in the present study, the underutilized softwood Scots pine (*Pinus sylvestris*) is fractionated using a novel low-cost PIL [N11H(2OH)][LAC]. The process parameters for the enhanced extraction of lignin are optimized. The biochemical composition of lignin and residual pulp was determined through a fiber analyzer. The surface morphology of the cellulose-rich pulp was characterized using SEM analysis. Further, the depolymerization properties of the extracted lignin were analyzed using GPC. The recovered lignin was characterized by ^1^H–^13^C HSQC NMR, quantitative ^31^P NMR, GPC, FTIR, and TGA analysis to determine the mechanism of fractionation from biomass.

## 2. Materials and Methods

### 2.1. Materials and Reagents

Prior to use, 2-dimethylamino-ethanol (>99.5%) was obtained from Sigma Aldrich (St. Louis, MO, USA) and; racemic lactic acid (>85%) was purchased from TCI Europe and used as received. Raw Scots pine (*Pinus sylvestris*) softwood biomass was collected from southern Estonia (58°09′28.4″ N 26°44′27.1″ E). The moisture content of the biomass was determined using Kern MLS-50-3D moisture analyzer (Kern & Sohn GmbH, Balingen, Germany). Fiber content of the pine biomass was determined using Ankom 200 fiber analyzer (ANKOM Technology, Fairport, NY, USA), and composition was estimated following Hasanov et al. [13]. In the present study, the estimated compositional analysis of pine wood biomass (% of dry mass) is cellulose, 47.93 ± 0.90; hemicellulose, 13.42 ± 0.76; lignin, 27.31 ± 0.53; ash, 3.40 ± 0.80; extractives, 4.12 ± 0.71; and moisture 3.36 ± 0.46. All the heating experiments were conducted in a conventional heating oven. 

### 2.2. Synthesis of PIL 2-Hydroxyethyl Dimethyl Ammonium Lactate [N11H(2OH)][LAC]

The protic ionic liquid 2-hydroxyethyl dimethyl ammonium lactate [N11H(2OH)][LAC] was synthesized by an acid-base neutralization method (Figure 1).

Briefly, 226 mL of 2-dimethylamino-ethanol (200 g, 1.05 equivalent to 2.24 mol) was added dropwise to a solution of 159 mL racemic lactic acid (192 g, 1.00 equivalent to 2.24 mol) and incubated in a water bath for 2 h. After entirely adding the base, the reaction continued for 6 h under constant stirring—the excess of amine produced during the neutralization process was eliminated using a high vacuum. After drying under a vacuum for 16 h, the product was obtained in quantitative yield as a slightly yellow, viscous liquid. The ^1^H and ^13^C NMR spectra of [N11H(2OH)][LAC] was recorded on a Bruker 400 MHz Avance III NMR (Billerica, MA, USA) spectrophotometer.

^1^H NMR (400 MHz, d6-DMSO): δ/ppm = 6.23 (s, 2H, CHOH + N-H), 3.87 (q, ^3^*J*_HH_ = 6.8 Hz, 1H, CHOH-CH_3_), 3.63 (t, ^3^*J*_HH_ = 5.6 Hz, 2H, N-CH_2_-CH_2_), 2.84 (t, ^3^*J*_HH_ = 5.6 Hz, 2H, N-CH_2_), 2.55 (s, 6H, N-CH_3_), 1.18 (d, ^3^*J*_HH_ = 6.8 Hz, 3H, CHOH-CH_3_).

^13^C[^1^H] NMR (101 MHz, d6-DMSO) δ /ppm = 178.55 (s, COOH), 66.73 (CHOH-CH_3_), 59.58 (s, N-CH_2_-CH_2_), 56.74 (s, N-CH_2_), 43.50 (s, N-CH_3_), 21.17 (CHOH-*C*H_3_).

### 2.3. [N11H(2OH)][LAC]-Mediated Biomass Fractionation

The collected pinewood was debarked, air-dried, ground to 1–2 mm, maintained at <10% moisture, and stored in an air-tight container. The fractionation and extraction of lignin from PWB using [N11H(2OH)][LAC] were adapted according to the methodology of Gschwend et al. [14]. Briefly, in a 100 mL ACE pressure tube with a silicone ring (front), PILs [N11H(2OH)][LAC] with various biomass-to-solvent ratios were added, vortexed, and incubated in a preheated oven. The samples were treated at different temperatures (150–210 °C), incubation times (0.5–4 h), and BM:PIL ratios (1:2, 1:3, 1:5). After the pretreatment, ethanol was added to separate the cellulosic pulp from PIL–lignin mixture by centrifugation at 4000 rpm for 10 min (3×), and the remaining ethanol was removed and recycled using a rotary vacuum evaporator (Büchi Rotavapor R-200, Büchi, Switzerland). Finally, centrifugation at 4000 rpm for 10 min (3×) was used to separate the lignin from the PIL using water as an anti-solvent.

### 2.4. Delignification and Lignin Recovery

The delignification of the PWB by [N11H(2OH)][LAC] was determined by estimating the residual lignin in the biomass before and after pretreatment on a dry weight basis [15]. Thus, the delignification percentage was calculated based on the equation given below:$$\%\;Delignification\;\left(\frac{w}{w}\right)=\left[\frac{(Lignin_{Raw\;Biomass)}-(Lignin_{PIL\;pretreated\;biomass)}}{Lignin_{Raw\;biomass}}\right]\times 100$$

The lignin recovery from [N11H(2OH)][LAC] pretreatment was determined by the amount of lignin retrieved from the precipitation relative to the initial lignin content in the raw PWB [16]. The lignin recovery was calculated on a dry weight basis using the following equation:$$\%\;Lignin\;recovery\;\left(\frac{w}{w}\right)=\left[\frac{Lignin_{Recovered\;from\;PIL}}{Lignin_{Raw\;biomass}}\right]\times 100$$

### 2.5. Scanning Electron Microscopy

SEM images of the biomass pretreated with [N11H(2OH)][LAC] and the untreated pine wood biomass were obtained using SEM Helios NanoLab 650 (FEI Company, Hillsboro, OR, USA) at the acceleration voltage of 10 KeV, and the images were obtained at various magnifications.

### 2.6. Characterization of Recovered Lignin

#### 2.6.1. ATR FT-IR

The surface functionalities of the lignin obtained from the [N11H(2OH)][LAC] pretreatment were measured using FTIR spectroscopy (Spectrum BXII, Perkin Elmer Inc., Waltham, MA, USA) with the universal attenuated total reflection (ATR) method. The spectra were recorded with an average accumulation of 16 scans in the 4000–600 cm^−1^ interval range with a resolution of 4 cm^−1^.

#### 2.6.2. Thermogravimetric Analysis

The thermal degradation properties of the [N11H(2OH)][LAC]-extracted lignin were determined using a NETZSCH STA 449 F3 Jupiter simultaneous (TGA and DSC/DTG) thermal analyzer (NETZSCH, Selb, Germany). Approximately 5 mg of the extracted lignin was heated up to 900 °C under N_2_ gas with a flow rate of 100 mL/min at a heating rate of 10 °C /min [17].

#### 2.6.3. Gel Permeation Chromatography

The molecular weight distribution of the [N11H(2OH)][LAC]-extracted lignin was analyzed using high-performance liquid chromatography (HPLC) (Shimadzu Prominence-i, LC-2030C 3D Plus, Shimadzu Corporation, Kyoto, Japan) equipped with LabSolutions GPC software. The HPLC system contained a pump, an auto-sampler, a set of 2 MCX columns (1000 Å and 100,000 Å), and a pre-column (8 mm × 50 mm) (Polymer Standards Service (PSS), GmbH, Mainz, Germany) with a UV detector (280 nm). The lignin samples were dissolved in 0.1 M NaOH (5 mg/mL), and isocratic flow was maintained with 0.1 M NaOH solution at a flow rate of 0.5 mL/min with an injection volume of 20 µL. The relative molecular weight of the lignin was determined using polystyrene sulfonate sodium salt standards (PSS, GmbH, Mainz, Germany) ranging from 1100 Da to 100,000 Da.

#### 2.6.4. NMR

The characterizations of the functional groups present in the [N11H(2OH)][LAC]-extracted PWB lignin were determined by ^31^P NMR and HSQC spectra. All of the NMR measurements (HSQC, quantitative ^13^C, and ^31^P NMR spectra) for the determination of the subunit abundance, interunit linkages, and phenolic OH content of the lignin were recorded on an Ascend Neo 500 MHz NMR spectrometer (Billerica, MA, USA)) equipped with a TCI Prodigy cryo-probe head following the procedure of Wang et al. [18].

## 3. Results and Discussion

### 3.1. Optimization of PWB Fractionation Using [N11H(2OH)][LAC]

The effects of the operational process parameters viz., the incubation time (0.5–4 h), temperature (150–210 °C), and BM:PIL ratio (1:2, 1:3, and 1:5) on PWB fractionation were investigated to determine the optimal operational conditions for maximal lignin extraction. The percentage of lignin extracted from the biomass, along with those of cellulose and hemicellulose, were determined by the compositional analysis of the recovered biomass obtained after the [N11H(2OH)][LAC] pretreatment, which was related to the biomass composition of untreated (UT) biomass (Figure 1).

Initially, the effect of the incubation time on lignin removal was studied by varying the incubation time and keeping the other parameters, i.e., the temperature and BM:PIL ratio, constant (Figure 1). The initial incubation time significantly improved lignin and hemicellulose removal. At the same time, the cellulose level was equivalent to that of the control. Lignin removal was found to be significant at 3 h, accounting for 33% delignification with a recovery of 27% (Figure 2).

After 3 h the delignification was gradually reduced, with a significant reduction in cellulose and hemicellulose content, possibly due to the acidic nature of PIL [16]. Further increasing the incubation time beyond the optimal level also affected the lignin recovery rate. Previous reports on the PIL-mediated delignification required a prolonged incubation period. The pretreatment of miscanthus with PIL [TEA][HSO_4_] required an optimal incubation period of 8 h [19]. Similarly, Chambon et al. [20] also used an 8 h incubation time for the pretreatment of South African sugarcane bagasse using PIL [TEA][HSO4]. Compared with the early reports, [N11H(2OH)][LAC] requires a shorter pretreatment period (3 h) for the efficient delignification and recovery of lignin from PWB.

Incubation temperature also plays a significant role in delignification and lignin recovery from biomass [21]. Thus, the delignification of PWB at three different temperatures (150 °C, 180 °C, and 210 °C) for 3 h at a 1:3 BM:PIL ratio was investigated. We found that [N11H(2OH)][LAC] delignification and lignin recovery from PWB significantly increases with an increase in temperature. The maximum delignification (60%) and lignin extraction (56%) by [N11H(2OH)][LAC] is achieved at 180 °C. This is concurrent with an earlier report about the requirement for higher temperatures and increased pretreatment times to complete the improved delignification of softwood biomass pine [12]. The occurrence of this phenomenon is due to the reduced viscosity of [N11H(2OH)][LAC] at elevated temperatures, which further increases the mixing and improves the mass-transfer properties. Additionally, acid attack on the ether linkages by the ammonium proton in [N11H(2OH)][LAC] is involved in the breakage of the *β–O–4*’ aryl ether bond, which is engaged in the simultaneous delignification and depolymerization of lignin [22].

The economic sustainability of PIL-mediated biomass pretreatment predominantly depends on a higher biomass loading ratio [16]. In the present study, the delignification was carried out at 180 °C for 3 h for all biomass loading ratios. From Figure 1 and Figure 2, it is evident that the optimum biomass loading for efficient lignin extraction is 1:3, where [N11H(2OH)][LAC] is able to delignify 60.16% PWB biomass with the extracted lignin yield of 56.14%. Biomass loading ratios of 1:2 and 1:5 also obtain similar rates of delignification, possibly because the efficiency of [N11H(2OH)][LAC] at 180 °C for 3 h has significantly enhanced the mass transfer properties at a reduced viscosity [23]. However, the improved cellulose content at the biomass loading ratio of 1:5 is probably due to the increased biomass wetting in [N11H(2OH)][LAC] [12]. Under the optimal process conditions (180 °C; 3 h; 1:3 BM: PIL ratio), [N11H(2OH)][LAC] exhibits 60.91% of delignification with a lignin recovery of 56%; the cellulose content of the recovered pulp is about 45%.

### 3.2. Scanning Electron Microscopy of [N11H(2OH)][LAC]-Delignified Pulp

The surface morphology of PWB before and after pretreatment using [N11H(2OH)][LAC] was visualized using SEM (Figure 3). The untreated PWB shows a highly ordered intact structure with a smooth surface displaying the compact three-dimensional fiber network of the original biomass (Figure 3a). After [N11H(2OH)][LAC] pretreatment, the recovered delignified pulp shows ruptured structures with more apertures on the surface of the biomass (Figure 3b).

The visible deformities on the surface can be attributed to the removal of lignin and hemicellulose from PWB [21]. Further, the effective infiltration of [N11H(2OH)][LAC] also reduces cellulose crystallinity and significantly disrupts the cell wall network of PWB [24]. Thus, the current study agrees with the earlier reported morphologies of PIL-mediated biomass pretreatment [25,26].

### 3.3. Characterization

#### 3.3.1. ATR FT-IR

The ATR-FTIR analysis of the [N11H(2OH)][LAC]-extracted PWB lignin under optimal conditions was compared with the commercial Kraft lignin to determine the changes in surface functional properties. The ATR-FTIR analysis results of the [N11H(2OH)][LAC]-extracted lignin and Kraft alkali lignin are shown in Figure 4.

Compared with the Kraft lignin, the [N11H(2OH)][LAC]-extracted PWB lignin displays improved surface properties due to the simultaneous extraction and functionalization of lignin by pretreatment. The characteristic band at 3439 cm^−1^ corresponds to the O=H stretching in the hydroxyl group of phenolic and aliphatic groups of lignin [22]. The peak at 2930 cm^−1^ indicates the C–H stretching in methyl and methylene groups. The detection of a new peak at 1688 cm^−1^ represents the carbonyl (C=O) group, which confirms the activation of PWB lignin during the [N11H(2OH)][LAC] pretreatment [27]. Further, the stretches of CO and CC of aromatic moieties of lignin were observed at 1594 cm^−1^ and 1515 cm^−1^, respectively [28]. The appearance of bands at 1375 cm^−1^ and 1275 cm^−1^ is assigned to the presence of syringyl and guaiacyl units, respectively [29]. Further, the peak at 824 cm^−1^ represents the plane C-H vibration of guaiacyl units [30].

#### 3.3.2. Thermogravimetric Analysis

Thermogravimetric analysis is used to characterize the thermal properties, such as stability and pyrolytic decomposition, at various temperatures, and to identify the potential relationship between degradation and chemical structure. Thus, the thermal stability of [N11H(2OH)][LAC]-extracted PWB lignin was determined using thermogravimetry (TGA) and derivative thermogravimetry (DTG), as shown in Figure 5.

The degradation of the [N11H(2OH)][LAC]-extracted lignin was divided into three stages. In the first stage, the initial weight loss observed at 120.6 °C is due to the evaporation of absorbed moisture [31]. The second degradation occurs between 180–450 °C, and is predominantly due to the cleavage of interunit linkages to volatile substances such as CO, CO_2,_ and CH_4_ [32]. The last phase is related to char combustion, and includes the decomposition of aromatic rings in methoxylation reaction of hydroxyl and volatile compounds from lignin [33]. From DTG, enhanced lignin decomposition occurs between the temperature range of 220 °C and 500 °C, and the maximum lignin degradation (DTG_max_) occurs at 263 °C, which is significantly lower than the commercial Kraft lignin (355 °C) [31]. The relatively lower thermostability is due to the higher fragmentation of lignin during [N11H(2OH)][LAC] extraction and the presence of weak linkages, which is concomitant with the earlier reports [33,34]. Further, the formation of highly condensed aromatic structures at elevated temperatures yielded a residual carbon content of 23% [29].

#### 3.3.3. Molecular Weight Distribution of [N11H(2OH)][LAC]-Extracted Lignin

The development of a high-value product from lignin is significantly influenced by the molecular weight and the polydispersity index (PDI) properties. Therefore, tuning these properties improves the application of lignin in various fields [35]. Thus, the molecular weight distribution of [N11H(2OH)][LAC]-extracted lignin from PWB, along with that of commercial alkali Kraft lignin, is presented in Table 1 and Figure 6.

From Table 1, it is evident that the lignin extracted using [N11H(2OH)][LAC] under optimal pretreatment conditions yielded a reduced weight average molecular weight (Mw) of 2208 g mol^−1^ and PDI of 3.28 from PWB compared to Kraft lignin (control). The decreased M_w_ and PDI of PWB obtained after pretreatment confirms the lignin depolymerization through breakage of interunit linkages [24]. Further, Figure 6 reconfirms the depolymerization of lignin by the formation of lower molecular weight peaks than alkali Kraft lignin during [N11H(2OH)][LAC] fractionation of PWB. The generation of low molecular weight units demonstrates their availability for the bioconversion of higher-valued products [7].

#### 3.3.4. HSQC and ^31^P NMR

The determination of structural units, the estimation of subunit composition, and the interunit linkages of [N11H(2OH)][LAC]-extracted PWB lignin were characterized by combining the two-dimensional heteronuclear quantum coherence (HSQC) NMR and quantitative ^13^C NMR (Appendix A). The assignments of structural units detected from HSQC were compared with the NMR database of cell wall model compounds [19]. The Scots pine (*Pinus sylvestris*) softwood biomass used in the present study predominantly constitute guaiacyl (G) units as major subunits (95%), with the major interunit linkage of *β–O–4*′ (45–50/100 aromatic units) [12]. Figure 2 represents the major subunits and interunit linkages of [N11H(2OH)][LAC]-extracted PWB lignin, and the numerical values for the volume integrals are present in Table 2.

The aromatic region of the HSQC for [N11H(2OH)][LAC]-extracted PWB lignin after 3 h at 180 °C depicts the characteristic peaks for phenylpropanoids viz., guaiacyl (G) and syringyl (S) units. The correlations for C_2_,C_5,_ and C_6_ represent G units at 110.9/6.91, 114.9/6.86, and 119.2/6.77, respectively, whereas the S units showed a prominent signal at S_2,6_ with a correlation at 104.1/6.63. The complete cleavage of the *β–O–*′ interunit linkage represents in situ depolymerization of PWB lignin during [N11H(2OH)][LAC] pretreatment [21]. Further, the lower abundance of other interunit linkages viz., phenylcoumaran (*β–′5′*) and resinol (*β–β′*)^α,γ^ indicates that these linkages were also chemically modified during [N11H(2OH)][LAC] pretreatment. Further, the occurrence of stilbene moieties [(*β–′5′*)^β^ and (*β–′1′*)^α^] due to the reverse aldol reaction of *β–′5′* displays modification of interunit linkages of recovered PWB lignin [36]. The accumulation of aryl-glycerol in the recovered PWB lignin corresponds to the breakage of S units from *β–O–4′* motifs [18]. The amount of benzyl ether linkage in pine is 2.2–2.5 per 100 monolignol units; however, the lower abundance of this linkage in HSQC NMR represents the selective hydrolysis of the lignin–carbohydrate complex by [N11H(2OH)][LAC] pretreatment [19,37]. The previous report about lignin extraction from pine wood using low-cost protic ionic liquid at an elevated temperature (170 °C) generated highly condensed lignin, evidenced by a higher G_2 cond_ signal [12]. However, the [N11H(2OH)[LAC]-mediated pretreatment of pine biomass at 180 °C for the 3 h did not produce signal intensities for the formation of condensed lignin in G or S subunits. Thus, the absence of peaks related to the traces of [N11H(2OH)][LAC], carbohydrates, and lignin condensation displays the efficiency of the pretreatment in selective fractionation of lignin from PWB.

The content of various hydroxyl groups, including aliphatic, phenolic, and carboxylic groups of [N11H(2OH)][LAC]-extracted lignin, is determined by ^31^P NMR (Appendix A) [38]. After [N11H(2OH)][LAC] pretreatment at the optimal conditions, ^31^P NMR showed an increase in the abundance of phenolic hydroxyl groups due to the cleavage of *β–O–4* ether linkage to form free phenolic alcohol end groups [19]. The primary composition of guiacyl subunits connected through *β-5* linkages contributes to higher guiacyl hydroxyl contents than that of C_5_-substituted hydroxyl (mostly with syringyl groups) in PWB [39]. The comparatively lower aliphatic hydroxyl groups in PWB are due to the dehydration reaction before the cleavage of the *β-O-4* linkage [40]. The relatively increased content of total phenolic hydroxyl groups demonstrates that [N11H(2OH)][LAC] is involved in the depolymerization of PWB lignin via the complete removal of *β–O–4* ether linkages. Further, the cleavage of the *β–O–4′* linkages with increased phenolic hydroxyl groups could produce depolymerized low-molecular-weight aromatic lignin units, which can be effectively transformed into higher-value products [13,41].

## 4. Conclusions

This study aimed at the synthesis of a novel PIL, 2-hydroxyethyl dimethyl ammonium lactate [N11H(2OH)][LAC], for the extraction of lignin from Scots pine biomass. Under optimized process conditions, [N11H(2OH)][LAC] recovered 56.13% of lignin from PWB. The compositional and SEM analyses of the recovered pulp showed selective removals of lignin. Further, the recovered lignin analyzed through ^1^H–^13^C HSQC NMR and quantitative ^31^P NMR reveals the cleavage of β–O–4′ linkages and a significant increase in hydroxyl groups. Moreover, the in situ depolymerization of [N11H(2OH)][LAC]-treated lignin was confirmed by TGA and GPC analyses. This implies that [N11H(2OH)][LAC] pretreatment is able to convert the recalcitrant softwood biomass (lignin) into depolymerized aromatic units, which can be extended for other softwood biomasses for value-added chemical production.

## Data Availability

Not applicable.

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
