# Peer review of "Efficient Lignin Fractionation from Scots Pine (Pinus sylvestris) Using Ammonium-Based Protic Ionic Liquid: Process Optimization and Characterization of Recovered Lignin"

_polymers, 2022, doi:10.3390/polym14214637_

Round 1

Reviewer 1 Report

This paper, entitled ‘Efficient Lignin Fractionation from Scots Pine (Pinus Syl-2 vestris) Using Ammonium-based Protic Ionic Liquid: Process Optimization and Characterization of Recovered Lignin’, is a scholarly work and can increase knowledge on this domain. The content is relevant to the Polymer journal and the authors provide an interesting and original study. The abstract and keywords are meaningful, the manuscript is very well written and well related to existing literature. I have few suggestions;

Authors wrote that they observed some sort of lignin depolymerization according to GPC results. I suggest authors to provide the total organic carbon (TOC) content of the water (aqueous) fraction? Is it possible to provide GCMS data to see which monomers if any are obtained? Can they also add more information on degree of depolymerization of recovered lignin?

Besides this have authors examined the possibility of recovering the protic ionic liquid afterwards? Is it possible to recycle the ionic liquid and how many times it can be recycled?

Finally, I also recommend authors to include 2D-HSQC and 31P NMR spectra in the manuscript or as supporting information.

I suggest that the article can be published after minor revisions.

Reviewer 3 Report

The submitted manuscript reports on the synthesis and application of protic IL for the fractionation and extraction of lignin from Scots pine softwood biomass. The manuscript is well organised. Objectives are clearly stated and an appropriate, brief background is provided. The results are good and adequate to show the potential of PILs to serve as medium for biomass processing. The conclusions are within the realm of presented data. However, the comparison with already existing literature seems rather short. More detailed comparing and contrasting already existing results would improve the overall impact and give perspective. Authors do not even attempt to compare the values obtained in this work with the results obtained in their previous work where another three PILs were investigated for the extraction of lignin from ash tree hardwood (ref 2). Also, the way of presentation needs to be improved prior to publication in order to meet the journal standards. This is primarily due to undeveloped writing skills (the introduction seems to be an exception). Also: -                 page 1, line 44, reference [2] – I find it more appropriate to cite the original report, where the data is evaluated, instead of simply siting the previous work of the authors -                 Scheme 1 of poor quality, please correct -                 Page 3, line 123: since various PWB:IL ratios have been investigated, in the fractionation procedure I would suggest to use generalised description (i.e., known/particular amounts) instead of giving the exact values of 5 and 15 g; also, it’s quite obvious that 5 to 15 is a 1:3 ratio…same on page 7, lines 221-222 - – it’s either 1:X (w/w) ratio or X% of biomass mass loading (no need for the same information to be given twice) -                 Page 3, line 125, Figure 2. caption, page 7, line 221, page 7 line 231, and so on (!!!)- it’s biomass to PIL, not PIL to biomass ratio, please verify throughout all the manuscript -                 Figure 2. Caption: ionosolv (???) retreatment -                 Page 7, line 247-248 – reference missing: which “earlier reported morphologies”? -                 Section 3.3. is missing subsections for particular characterisation methods, i.e., 3.3.1. ATR FT-IR, 3.3.2. Thermogravimetric analysis, etc -                 Figure 6 – IL name with “11” in subscript all of the sudden -                 Table 2. is very messy and difficult to read, please correct; also, in the caption: “… NMR, HSQC, and analysis” (???) -                 Section 4., first sentence: 2-hydroxyethyl ammonium lactate OR (?) [N11H(2OH)][LAC]… -                 “Author contribution” section (page, 13, lines from 378) needs to be filled properly -                 Reference [13] is missing the “issue” number: 114 -                 Some typos: page 3, line 113: some swirl sign instead of “δ”, page 3, line 118: Section no: 2.3.[.; page 3, line 125: PIL: BR ratio (vs. BM on page 4, line 180)
